# Safety and Efficacy of Irreversible Electroporation in Locally Advanced Pancreatic Cancer: An Evaluation from a Surgeon’s Perspective

**DOI:** 10.3390/cancers14225677

**Published:** 2022-11-18

**Authors:** Jian Shen, Penglin Pan, Xiaoli Hu, Jun Zhao, Heshui Wu

**Affiliations:** 1Department of Pancreatic Surgery, Union Hospital, Tongji Medical College, Huazhong University of Science and Technology, Wuhan 430022, China; 2Department of Radiology, Wuhan Asia Heart Hospital, Wuhan 430022, China; 3Department of Anatomy, School of Basic Medicine, Huazhong University of Science and Technology, Wuhan 430030, China

**Keywords:** irreversible electroporation, pancreatic cancer, complication, survival, outcome

## Abstract

**Simple Summary:**

Pancreatic cancer is the fourth most lethal human malignancy. One-third of pancreatic cancer cases are diagnosed as locally advanced pancreatic cancer (LAPC); however, the optimal treatment of LAPC remains to be elucidated. Irreversible electroporation (IRE) has been applied as the treatment LAPC, but the safety and efficacy of IRE against LAPC need to be further clarified. In this study, we evaluate the safety and efficacy of IRE against LAPC, as well as exploring its impact on the immune response. The rate of major complications in the IRE group was similar to that in those undergoing pancreaticoduodenectomy with concurrent vascular resection (VR group), but higher than patients undergoing palliative surgery (PS group). The overall survival of the IRE group was shorter than that of the VR group, but longer than that of the PS group. The survival advantage in IRE-treated patients may be attributed to tumor ablation and immune modulation effects. Therefore, IRE is a feasible treatment for patients with LAPC.

**Abstract:**

Irreversible electroporation (IRE) has emerged as a promising treatment for patients with locally advanced pancreatic cancer (LAPC). Therefore, in this study, we evaluate the safety and efficacy of IRE against LAPC, as well as exploring its impact on anti-tumor immunity. A retrospective analysis was conducted in consecutive patients at a single institution. Eligible patients were assigned to IRE, palliative surgery (PS), or vascular resection (VR) groups, according to their respective treatments. The IRE group consisted of LAPC patients. One-to-one propensity score matching was performed, in order to compare the incidence of complications and median overall survival (mOS). Serum and intratumoral cytokines, as well as intratumoral immune cells, were analyzed in order to identify changes in immunity after IRE. A total of 210 patients were included. After matching, the rate of major complications (Clavien–Dindo III–V), intra-abdominal hemorrhage, and re-intervention in the IRE group were similar to those in the VR group (*p* > 0.05). The mOS of the IRE group (13.0 months) was shorter than that of the VR group (15.0 months), but longer than that of the PS group (8.0 months) (*p* < 0.05). Patients in the IRE group had elevated serum levels of immunogenic cytokines, including IL-2, IL-6, and TNF-α, which were related to anti-tumor immunity. The survival advantage in IRE-treated patients was attributed to tumor ablation and immune modulation effects. Overall, IRE can be considered a feasible treatment for patients with LAPC.

## 1. Introduction

Pancreatic cancer is one of the most lethal types of human malignancies, with almost equal incidence and mortality rates. It is the 14th most common cancer and the 7th most common cause of cancer death around the globe [1,2]. Early diagnosis of pancreatic cancer is hindered by a lack of specific early clinical symptoms. As a result, a majority of patients are diagnosed at an advanced stage or with distant metastases, and more than 430,000 patients succumb to this disease per year [1,2,3]. While surgical resection is the only curative therapy for pancreatic cancer, only 20% of patients are suitable for this treatment [3,4,5]. Despite the improvements in medical care over the past few decades, the prognosis of patients with pancreatic cancer remains abysmal, with a 5-year survival rate below 7% [3,6].

Patients with locally advanced pancreatic cancer (LAPC) account for one-third of all pancreatic cancer cases. Ineligible for surgical resection, this cohort of patients receive systemic chemotherapy as the primary treatment, yet only with limited success [3,5,6,7,8,9]. Therefore, several novel local treatments are being investigated for their efficacies against LAPC. Thermal ablation, for example, is widely used to treat many types of solid tumors; however, it suffers from high rates of morbidity and mortality in patients with PDAC [4,6]. Electroporation, on the other hand, is a non-thermal ablation technique that delivers short but high-intensity electric pulses to produce nanoscale pores in the cytoplasmic membranes of cancer cells, subsequently inducing cell death. Electroporation is advantageous over other conventional thermal ablation methods in many ways, including minimal damage to blood vessels or bile ducts, as well as retaining the regenerative potential of treated tissues. According to its working mechanism, electroporation can be divided into reversible electroporation (RE) and irreversible electroporation (IRE). RE transiently increases cell permeability without directly killing the tumor, facilitating cellular cytotoxic drug uptake in tumor cells. Therefore, RE is typically used in combination with chemotherapeutic drugs as a treatment for tumors—this treatment is called electrochemotherapy (ECT) [10,11,12,13,14,15]. Meanwhile, IRE uses a higher electric field and number of pulses to cause cell death by inducing irreversible cell membrane perforation. As a radical ablative therapy, IRE has been widely used for the treatment of soft-tissue tumors, such as pancreatic cancer, liver cancer, kidney cancer, and prostate cancer. Both ECT and IRE have been used clinically to treat human cancers [7,8,9,10,11,12,13,14,15,16,17,18]. IRE can kill tumors completely, with well-demarcated borders and tolerance of the heat-sink effect [6,18]. As a radical ablative therapy, IRE was used in the treatment of soft-tissue tumors, such as pancreatic cancer, liver cancer, kidney cancer, and prostate cancer. Nanoknife, a commercialized piece of equipment designed for IRE, was first used to treat patients with LAPC in 2009, and it has thus far proven efficacious in many clinical trials. As a result, it was approved by the Food and Drug Administration of the United States in 2012, and by the National Medical Products Administration in 2015 [7,18]. Although studies have shown that ECT is a safe treatment for tumors, IRE is a clinically approved treatment for LAPC [9,10,11,12,13,14,15,16,17,18,19].

Reports on the prognosis of IRE have accrued as it gains popularity in treating LAPC. However, most of these reports have only described the outcomes of IRE itself, and fall short in comparing IRE with other treatment modalities [9]. IRE is known to modulate the tumor micro-environment and can potentially improve the prognosis of patients. However, most related evidence is based on animal models [20,21]. In this study, we aimed to investigate the safety and long-term outcomes associated with IRE, through a comparison of surgeries in a real-world setting. We also analyzed the possible causes for the improved efficacy of IRE from the perspective of immune responses.

## 2. Materials and Methods

### 2.1. Study Design

A retrospective cohort study at a single institution was conducted to analyze the safety and long-term outcomes of IRE. Tumor resectability was determined by pre-operative multi-phase contrast-enhanced computed tomography. Eligible patients were then allocated into three groups: patients with LAPC at the pancreatic head or neck treated with IRE (IRE group), patients undergoing palliative surgery (PS group), and patients receiving pancreaticoduodenectomy with concurrent vascular resection (VR group). Given that LAPC patients are unsuitable for surgical resection, we used the VR group as a positive control group and PS serve as negative control, respectively.

### 2.2. Patient Population

The electronic medical records of LAPC patients treated with IRE between 2017 and 2019 at a single institution were retrospectively reviewed. Data from patients in the PS and VR groups were also collected.

Patients were excluded if they met one of the following criteria: disease complicated by chronic organ insufficiency (defined based on the presence of organ insufficiency for more than 3 months, involving the malfunction of heart, liver, kidney, lung, coagulation, or circulation), age younger than 18 years, tumor located at the body and tail of the pancreas, pathologically verified non-pancreatic cancer, and the presence of metastasis. In addition, in the VR group, patients who received surgery combined with total pancreatectomy or duodenum-preserving pancreatic head resection were excluded. Patients that met the following criteria were treated with IRE: LAPC patients with tumors less than 5 cm in diameter, no metal stent or myocardial infarction, and no portal vein stenosis. Patients were included in the PS group if they were found unsuitable for resection, but did not present with metastasis.

### 2.3. Data Collection and Definition

The primary endpoints of this study were post-operative complications and survival. Data on patient demographics, medical history, biochemical markers, intraoperative variables, and post-operative complications were collected from the medical records. Data from the patients discharged from the hospital were collected through outpatient visits, telephone calls, or internet messages. The following complications were included: postoperative fistula, re-admission, intra-abdominal hemorrhage, pneumonia, re-intervention, intra-abdominal infection, re-operation, and delayed gastric emptying. The Clavien–Dindo classification (CDC) was used to stratify operative complications.

LAPC was defined according to the criteria of the NCCN guidelines of pancreatic adenocarcinoma: greater than 180° involvement of the celiac axis or superior mesenteric artery by the solid tumor, or an un-reconstructable portal vein or superior mesenteric vein, and no evidence of metastasis. The length of tumor–vessel contact was measured from preoperative computed tomography images. Pancreatic fistula, intra-abdominal hemorrhage, and delayed gastric emptying were classified according to the 2016, 2007, and 2007 International Study Group of Pancreatic Surgery guidelines, respectively. The definition of pancreatic fistula, intra-abdominal hemorrhage, delayed gastric emptying re-intervention, re-operation, re-admission, overall survival, mortality, comprehensive complication index (CCI), and prognostic nutritional index (PNI) are presented in detail in Appendix A
Table A1.

### 2.4. Perioperative Management

Blood tests were routinely conducted within one week prior to the surgery. Clinical examination and pre-operative imaging were performed to verify the location and clinical stage of tumors, which were used for the optimization of therapeutic schedules.

Prophylactic antibiotics were routinely administered a half-hour before surgery. Surgical approaches were performed by surgeons with extensive experience in pancreatic surgery, in accordance with specific surgical principles and guidelines. All patients underwent open laparotomy under general anesthesia with muscle relaxants, and the surgical procedure was customized according to the location and size of the tumor, symptoms, and patient preferences. For patients treated with IRE, the Nanoknife system was used in situ without tumor resection. The number of electrodes was first selected based on the tumor size. Then, the electrodes were placed under ultrasound guidance, in order to achieve complete ablation of the tumor and to avoid damage to peritumoral vascular structures and the pancreatic duct. The following settings were employed: interelectrode distance, 15–20 mm; active tip length, 15 mm; field strength, 1500 V/cm; current, 30–40 A; pulse length, 90 ms; and pulses, 90. An electrode pullback was subsequently performed to treat the superficial part of the tumor. At the end of IRE, ultrasound was again used to confirm the complete coverage of tumor region by the hyperechoic area. Biliary–intestinal and gastrointestinal anastomoses were performed, according to the doctor’s choice. Patients in the PS group underwent biliary–intestinal or gastrointestinal anastomosis with or without celiac ganglion alcoholization. Patients in the VR group received pancreaticoduodenectomy combined with vascular resection, after which vascular reconstruction was performed. Child’s-type digestive tract reconstruction and end-to-side duct-to-mucosa pancreaticojejunostomy were applied in all patients who underwent resection. None of the patients in the VR group underwent margin accentuation with IRE. No additional biological materials were used, except for artificial blood vessels. Routine peritoneal drainage tubes were placed.

For gastric acid suppression, somatostatin was routinely administered, while its analogs were not. An early post-operative imaging scan was performed to evaluate acute complications. Additional tests or examinations were performed in the case where the occurrence of complications was suspected. Early oral intake, ambulation, and withdrawal of drainage tubes were recommended. Patients in good clinical condition but with a high concentration of amylase in drainage fluid were discharged with drainage tubes, which were removed during their follow-ups after fistula had disappeared.

### 2.5. Follow-Up

Patients received regular follow-ups after discharge. To assess the changes in their physical conditions, clinical examination was performed every three months in the first year, every four months in the following two years, and twice a year thereafter. Adjuvant therapy was prescribed according to the patient’s pathological results. The most-used adjuvant options included S-1 monotherapy or gemcitabine combined with albumin-bound paclitaxel.

### 2.6. Statistical Analysis

Continuous variables are presented as the means and standard deviations, medians, and interquartile ranges. Comparisons among variables were performed using one-way ANOVA or Kruskal–Wallis tests as appropriate. Categorical variables are expressed as absolute numbers or frequencies, and were analyzed using the χ^2^ or Fisher’s exact test. Pairwise comparisons were adjusted for multiple tests through Bonferroni correction. One-to-one propensity score matching with a matching tolerance of 0.2 was conducted, in order to avoid the introduction of potentially confounding factors, and scores were calculated through logistic regression analysis. Variables that were statistically different in the univariate analysis were included in propensity matching analysis. The mOS among the groups was assessed using the Kaplan–Meier method and compared using the log-rank test. Missing data were removed. Statistical analysis was performed using the SPSS 26.0 software (SPSS Inc., Chicago, IL, USA) and R software (version 4.1.3).

## 3. Results

### 3.1. Clinical Characteristics of the Patients

In total, 210 consecutive patients were enrolled, comprising 119 males and 91 females with a median age of 60.0 ± 9.1 years and an average BMI of 21.8 ± 2.6. Patients receiving neoadjuvant therapy were not enrolled, due to their scarcity in this cohort. The patients were assigned into three groups according to the surgical procedures, with 63 patients in the IRE group, 58 patients in the vascular reconstruction group (mainly reconstruction of the portal vein or superior mesenteric vein), and 89 patients in the palliative surgery group. IRE ablation was successfully performed in all cases. A total of 91.4% (53/58) of patients in the VR group underwent R0 resection. The characteristics of patients are summarized in Table 1.

### 3.2. Early Clinical Response

The VR group was included as the positive control regarding treatment safety, as VR is considered to have a high rate of complications. As shown in Table 1, significant differences in BMI, tumor size, PNI, and pre-operative albumin were observed among the groups, primarily in the PS group. After matching, each group included 58 cases, and no significant differences in clinical characteristics were noted among the groups.

The parameters of short-term outcomes are summarized in Table 2. In the IRE group, the most common complication after IRE treatment was delayed gastric emptying (12.7%), followed by intra-abdominal hemorrhage (7.9%), and intra-abdominal infection (3.2%). Two deaths (3.2%) were observed (one for septic shock and the other for intra-abdominal hemorrhage). A total of 17.5% of IRE-treated patients experienced CDC III–V complications, and 12.7% of them required reintervention. Differences were found in the rates of pancreatic fistula, intra-abdominal infection, reintervention, and CDC III–V complications among the groups before matching. The rates of pancreatic fistula and intra-abdominal infection were highest in the VR group. The percentages of re-intervention and CDC III–V complications in the IRE group were equal to those noted in the VR group, but higher than those noted in the PS group. After matching, the rates of pancreatic fistula and intra-abdominal infection in the VR group were higher than those in the other groups. Patients in the IRE group and VR group experienced more grade III–V complications, re-intervention, and intra-abdominal hemorrhage, compared with those in the PS group.

### 3.3. Long-Term Prognosis of Patients

Patient outcomes in the three groups were also evaluated, in order to further elucidate the efficacy of IRE. The median follow-up duration of all patients was 22.0 months (range: 1.0–32.0 months), while 32 patients (15.2%) were censored. Propensity score matching was performed based on BMI, pre-operative albumin, PNI, CA 19-9, tumor size, and length of tumor–vessel contact, after which each group consisted of 40 cases. The characteristics are summarized in Table A2. The survival of the IRE group (13.0 months, 95% CI: 10.9–15.1) was similar to that of the PS group (8.0 months, 95% CI: 6.6–9.4, *p* = 0.053), but short than that of the VR group (15.0 months, 95% CI: 13.5–16.5, *p* = 0.005). After matching, the OS (13.0 months, 95% CI: 11.0–15.0 of the IRE group was longer than that (8 months, 95% CI: 5.9–10.1) of the PS group (*p* = 0.008), but shorter than that (15.0 months, 95% CI: 11.5–16.5) of the VR group (*p* = 0.041). The survival curves are presented in detail in Figure 1.

### 3.4. Peripheral Immune Responses after IRE

In the study, 22.4% of patients in the PS group, 46.6% of patients in the IRE group, and 41.4% of patients in the VR group were tested for serum cytokines. As shown in Figure 2, there were no significant differences in the pre-operative serum levels of IL-2, IL-4, IL-6, IL-10, TNF-α, or INF-γ among the three groups, while elevation of these cytokines to varying extents was observed on day 1 after surgery. When compared to the PS group, the IRE group presented significantly elevated serum IL-2 (*p* = 0.035, Figure 2A), IL-6 (*p* = 0.043, Figure 2D), and TNF-α (*p* = 0.043, Figure 2E) on day 1 after surgery. The mean serum IL-2 and TNF-α levels in the IRE group were higher than those in the other groups (*p* < 0.05, Figure 2A,E). The IFN-γ level was higher on day 1 after surgery than before surgery, although there was no difference among the three treatment groups (*p* < 0.05, Figure 2F). GO enrichment analysis indicated that these elevated cytokines are associated with the immune response (Figure 2G). The neutrophil–lymphocyte ratio (NLR), an immune marker related to poor prognosis, was lower in the IRE group than in the PS group (Figure 2H).

## 4. Discussion

For patients with unresectable pancreatic tumors, electroporation may be a treatment option, as it can preserve the surrounding anatomical structures and, therefore, has gained popularity for treating patients with LAPC [4,5,6,10,11,12,13,14,15,16,17]. To the best of our knowledge, this is the first study comparing the clinical performance of IRE in LAPC with other surgical modalities. We showed that IRE yielded a significantly longer mOS than found in the PS group. Perioperative complications in the IRE group were also well-tolerated, and anti-tumor immunity was observed in the IRE group. Therefore, IRE potentially represents a feasible and effective treatment for LAPC.

The incidence and severity of post-operative complications are important factors that limit the clinical application of IRE. Unwanted thermal effects, unexpected healthy pancreatic tissue necrosis, and tissue edema may be the root causes of such complications. In the IRE group, 17.5% of patients experienced CDC III–V complications, and two deaths were observed. Compared to the VR group, the IRE group experienced lower incidences of pancreatic fistula and intra-abdominal infection. However, no difference was found in the rates of other complications between the groups. No significant difference was found in the rates of pancreatitis, re-operation, or death, which may be attributed to the limited sample size. In our study, the complication rates after IRE were comparable to those in other studies. A recent prospective study on Nanoknife has shown that 53.0% of patients with LAPC had an uneven treatment course and that 20.0% of patients experienced major complications (grade III and higher) after treatment, with a 30-day mortality rate of 5.0% [5]. Another multi-center prospective study has reported a major adverse event rate of 42% and a 90-day mortality rate of 4% [8]. A systematic review on the safety and efficacy of 304 patients treated with IRE concluded that the rate of severe complications was 0–24% and that the mortality rate was 0–17% [17]. In order to evaluate the safety of IRE against LAPC, the PS and VR groups were introduced in this study. As is well-known, the VR group is considered to have a high but acceptable complication rate [22]. The rate of complications in the IRE group was not higher than in the VR group. Therefore, we can draw the conclusion that IRE is a feasible treatment for patients with LAPC. We also attempted to explain the higher post-operative complications in the IRE group. As the imbalance of post-operative inflammatory change is an initiating factor for post-operative complications, we explored the post-operative changes in cytokines. IL-6, a dominating inflammatory cytokine, has presented higher levels in patients after abdominal surgeries, especially in those who developed post-operative complications [23]. In this study, a higher IL-6 level was observed in the IRE and VR groups, both of which presented more complications; thus, IL-6 was positively correlated with the severity of complications (Figure A1A,B). The AUC (0.87) for IL-6 in differentiating severe complications shows the superiority of serum IL-6 on post-operative day 1 (Figure A1C). Based on our results, IL-6 may be used to predict post-operative complications

In this study, the mOS in the IRE group was longer than in the PS group (13.0 months vs. 8.0 months, *p* < 0.05), highlighting the efficacy of IRE against LAPC. The outcomes reported for the IRE group were consistent with those described in previous reports. One multi-center prospective study has reported a mOS of 10.0 months (95% CI: 8.0–11.0 months) in LAPC patients undergoing IRE treatment [8], while another reported a median mOS of 10.7 months [5]. It is worth noting that better outcomes have also been reported in some clinical studies. One study in 200 LAPC patients reported a mOS of 23.2 months (range: 4.9–76.1) after IRE treatment, and a prospective multi-institution study reported a mOS of 30.7 months (range: 0.2–68.3 months) among 152 patients undergoing IRE [24]. Several factors may contribute to these striking findings. First, the patients in those studies were treated by chemotherapy prior to IRE, and patients with biologically aggressive disease were excluded [8]. Second, standard chemotherapeutic intervention was administered to most patients. In comparison, a majority of the patients in our study did not receive chemotherapy, considering their preferences. Nevertheless, we have shown that IRE provided survival benefits for patients with LAPC.

To clarify why patients can benefit from IRE treatment, the immune response after IRE was assessed. Cytokines are crucial to the anti-tumor immune response. Elevated levels of IL-2, IL-6, and TNF-α were noted compared with levels in the PS group. IL-2 is crucial for the proliferation of antigen-stimulated T lymphocytes and immune responses involving cytotoxicity [25], where reduced IL-2 has been reported to be correlated with poor prognosis. Due to its immunostimulatory properties, IL-2 has been treated as a therapeutic option for several tumors, and most IL-2-treated patients have experienced prolonged disease-free survival [26]. IL-6 is an essential factor for the proliferation and function of T lymphocytes. TNF-α is an acute response cytokine, the elevated level of which after treatment also coincides with previous reports. TNF-α is an immunostimulatory factor, which could create a pro-inflammatory environment. TNF-α could enhance the sensitivity of tumors to treatment and promote apoptosis in the tumor [27]. GO enrichment analysis of the elevated cytokines indicated that they were associated with immune response. NLR in the IRE group was lower than that in the PS group, suggesting that immune responses in peripheral blood are regulated by IRE (Figure 2H). The following reasons may account for the promising prognosis after IRE: IRE promotes cytoreduction, and IRE treatment shifts the immune micro-environment toward an anti-tumor phenotype [16,28]. Due to the limited ability of IRE to improve prognosis, it seems that the major effect of IRE is cytoreduction, and the pursuit of complete cytoreduction with an IRE system should be further explored. 

It should be noted that electrochemotherapy, i.e., reversible electroporation in combination with chemotherapy, has also shown efficacy against pancreatic cancer by increasing the tumor uptake of chemotherapeutic drugs [10,11,12,13,14]. In a recent clinical trial on patients with locally advanced pancreatic cancer, electrochemotherapy produced an overall survival of 11.5 months and rapidly resolved abdominal pain [10]. Therefore, it is hopeful that electrochemotherapy would be approved in the near future and further expand the toolbox for pancreatic cancer therapy.

Our study had several limitations. First, we conducted a retrospective study, although patient enrollment was carefully designed to overcome potential bias. We also attempted to adjust for confounders using statistical methods, such as propensity matching. Second, the sample size in this study was limited. Third, serum cytokines were only tested in a fraction of patients, as this is not a routine lab test. Fourth, the PS and VR groups were not ideal controls. Therefore, conclusions obtained from our study should only be cautiously extrapolated to other scenarios. In the future, a multi-center prospective study should be performed.

## 5. Conclusions

In conclusion, we evaluated the safety and efficacy of irreversible electroporation for locally advanced pancreatic cancer and investigated the immune activity induced by IRE in patients.

## Figures and Tables

**Figure 1 cancers-14-05677-f001:**
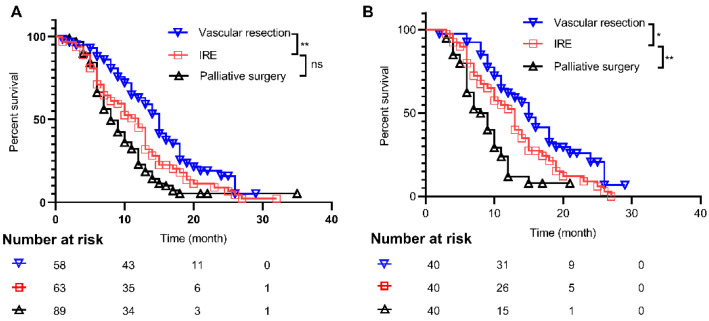
Kaplan–Meier curves of overall survival for patients in different groups: (**A**) Comparison of overall survival among the three groups before matching; and (**B**) comparison of overall survival among the three groups after matching. *, *p* < 0.05; **, *p* < 0.01; ns, not significant; IRE, irreversible electroporation.

**Figure 2 cancers-14-05677-f002:**
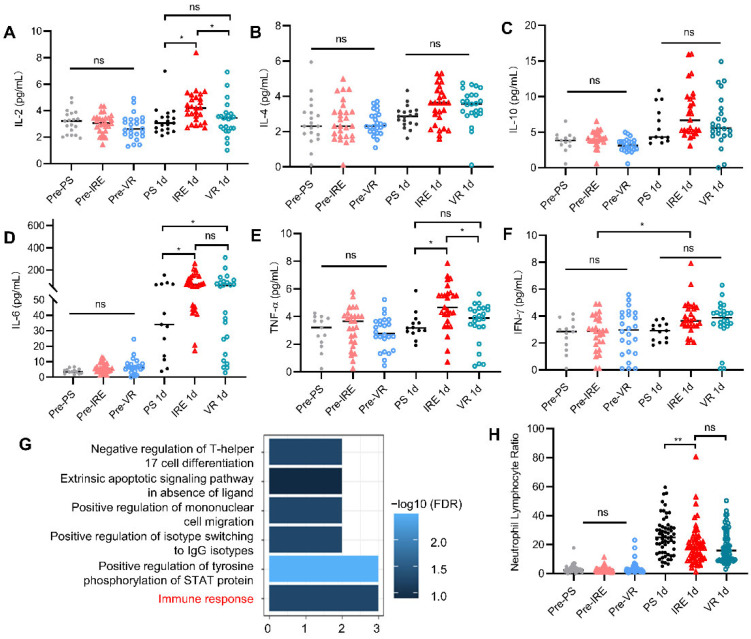
Changes in serum cytokine levels on the first day after treatment in patients treated with indicated procedure: (**A**) Changes in IL-2; (**B**) changes in IL-4; (**C**) changes in IL-10; (**D**) changes in IL-6; (**E**) changes in TNF-α; (**F**) changes in INF-γ; (**G**) GO enrichment analysis of the different cytokines; and (**H**) changes in neutrophil–lymphocyte ratio, an immune marker. *, *p* < 0.05; **, *p* < 0.01; ns, not significant.

**Table 1 cancers-14-05677-t001:** Demographics and clinical characteristics of the patients.

Variables	Before Matching				After Matching			
	IRE Group (*n* = 63)	Vascular Resection Group (*n* = 58)	Palliative Surgery Group (*n* = 89)	*p*	IRE Group (*n* = 58)	Vascular Resection Group (*n* = 58)	Palliative Surgery (*n* = 58)	*p*
Age (years)	60.0 ± 9.1	59.4 ± 8.7	61.0 ± 9.3	0.782	60.7 ± 9.0	59.4 ± 8.7	60.5 ± 9.1	0.969
BMI (kg/m^2^)	21.8 ± 2.5	22.3 ± 2.3	21.2 ± 2.9	0.022	21.7 ± 2.4	22.3 ± 2.3	21.4 ± 2.5	0.129
Sex				0.892				0.999
Female	26 (41.2%)	25 (43.1%)	40 (45.0%)		25 (43.1%)	25 (43.1%)	25 (43.1%)	
Male	37 (58.8%)	33 (56.9%)	49 (55.0%)		33 (56.9%)	33 (56.9%)	33 (56.9%)	
Smoking				0.763				0.718
Yes	14 (22.2%)	16 (27.6%)	24 (27.0%)		12 (20.7%)	16 (27.6%)	14 (24.1%)	
No	49 (77.8%)	42 (72.4%)	65 (73.0%)		46 (79.3%)	42 (72.4%)	44 (75.9%)	
Drinking				0.479				0.718
Yes	15 (23.8%)	12 (20.7%)	14 (15.7%)		15 (25.9%)	12 (20.7%)	11 (19.0%)	
No	48 (76.2%)	46 (79.3%)	75 (84.3%)		43 (74.1%)	46 (79.3%)	47 (81.0%)	
Diabetes				0.249				0.268
Yes	15 (23.8%)	7 (12.1%)	17 (19.1%)		14 (24.1%)	7 (12.1%)	11 (19.0%)	
No	48 (76.2%)	51 (87.9%)	72 (80.9%)		44 (75.9%)	51 (87.9%)	47 (81.0%)	
Hypertension				0.712				0.487
Yes	15 (23.8)	15 (25.9%)	18 (20.2%)		15 (25.9%)	15 (25.9%)	10 (17.2%)	
No	48 (76.2)	43 (74.1%)	71 (79.8%)		43 (74.1%)	43 (74.1%)	48 (82.8%)	
Tumor size (cm)	4.1 ± 1.3	3.7 ± 1.4	4.2 ± 1.6	0.027	3.9 ± 1.	3.7 ± 1.4	3.8 ± 0.9	0.125
CA 19-9 (U/L)	188.4 (34.2–808.3)	321.6 (35.5–1200.0)	645.6 (75.8–1200.0)	0.055	272.9 (37.1–1200.0)	331.1 (33.3–1200.0)	700.0 (84.2–1200.0)	0.152
CA 125 (U/L)	28.1 (15.4–52.1)	24.1 (11.4–47.3)	29.1 (16.9–56.1)	0.160	29.7 (15.4–50.6)	25.4 (13.9–44.8)	27.4 (16.6–47.1)	0.346
PNI	47.1 ± 6.3	46.3 ± 8.1	44.1 ± 7.5	0.002	47.0 ± 5.6	46.3 ± 8.1	45.4 ± 5.5	0.130
Pre-operative albumin	39.6 ± 5.3	38.9 ± 6.6	37.9 ± 5.0	0.024	39.6 ± 4.9	38.9 ± 6.6	39.1 ± 4.1	0.203
Total bilirubin (μmol/L)	17.4 (11.5–70.9)	20.8 (12.7–87.6)	49.6 (14.5–121.3)	0.028				
ASA				0.753				0.634
I–II	46 (73.0%)	40 (69.0%)	60 (67.4%)		42 (72.4%)	40 (69.0%)	37 (63.8%)	
III–IV	17 (27.0%)	18 (31.0%)	29 (32.6%)		16 (27.6%)	18 (31.0%)	21 (36.2%)	
Adjuvant chemotherapy				0.187				0.268
Yes	16 (25.4%)	11 (19.0%)	12 (13.5%)		14 (24.1%)	11 (19.0%)	7 (12.1%)	
No	47 (74.6%)	47 (81.0%)	77 (86.5%)		44 (75.9%)	47 (81.0%)	51 (87.9%)	

**Table 2 cancers-14-05677-t002:** Post-operative short-term outcomes of the patients.

Complications	Before Matching				After Matching			
	IRE Group (*n* = 63)	Vascular Resection Group (*n* = 58)	Palliative Surgery Group (*n* = 89)	*p*	IRE Group (*n* = 58)	Vascular Resection Group (*n* = 58)	Palliative Surgery Group (*n* = 58)	*p*
Pancreatic fistula				<0.001				0.001
Yes	1 (1.6%)	9 (15.5%) ^†^	0 (0.0%)		1 (1.7%)	9 (15.5%) ^†^	0 (0.0%)	
No	62 (98.4%)	49 (84.5%)	89 (100.0%)		57 (98.3%)	49 (84.5%)	58 (100.0%)	
Intra-abdominal hemorrhage				0.604				0.081
Yes	5 (7.9%)	5 (8.6%)	4 (4.5%)		5 (8.6%)	5 (8.6%)	0 (0.0%) ^†^	
No	58 (92.1%)	53 (91.4%)	85 (95.5%)		53 (91.4%)	53 (91.4%)	58 (100%)	
Intra-abdominal infection				0.025				0.035
Yes	2 (3.2%)	9 (15.5%) ^†^	5 (5.6%)		2 (3. 5%)	9 (15.5%) ^†^	3 (5.2%)	
No	61 (96.9%)	49 (84.5%)	84 (94.4%)		56 (96.5%)	49 (84.5%)	55 (94.8%)	
Delayed gastric emptying				0.408				0.569
Yes	8 (12.7%)	7 (12.1%)	17 (19.1%)		5 (8.6%)	7 (12.1%)	9 (15.5%)	
No	55 (87.3%)	51 (87.9%)	72 (80.9%)		53 (91.4%)	51 (87.9%)	49 (84.5%)	
Pancreatitis				0.474				0.999
Yes	1 (1.6%)	1 (1.7%)	0 (0.0%)		1 (1.7%)	1 (1.7%)	0 (0.0%)	
No	62 (98.4%)	57 (98.3%)	89 (100.0%)		57 (98.3%)	57 (98.3%)	58 (100%)	
Re-intervention				0.008				0.059
Yes	8 (12.7%)	7 (12.1%)	1 (1.1%) ^†^		8 (13.8%)	7 (12.1%)	1 (1.7%) ^†^	
No	55 (87.3%)	51 (87.9%)	88 (98.9%)		50 (86.2%)	51 (87.9%)	57 (98.3%)	
Re-operation				0.264				0.999
Yes	2 (3.2%)	1 (1.7%)	0 (0.0%)		1 (1.7%)	1 (1.7%)	0 (0.0%)	
No	61 (96.9%)	57 (98.3%)	89 (100.0%)		57 (98.3%)	57 (98.3%)	58 (100%)	
30-day re-admission				0.652				0.301
Yes	5 (7.9%)	3 (5.2%)	4 (4.5%)		5 (8.6%)	3 (5.2%)	1 (1.7%)	
No	58 (92.1%)	55 (94.8%)	85 (95.5%)		53 (91.4%)	55 (94.8%)	57 (98.3%)	
Clavien–Dindo classification				0.002				0.020
lower than grade III	52 (82.5%)	46 (79.3%)	86 (96.6%)		48 (82.8%)	46 (79.3%)	56 (96.6%)	
Grade III and higher	11 (17.5%)	12 (20.7%)	3 (3.4%)^†^		10 (17.2%)	12 (20.7%)	2 (3.4%) ^†^	
Death				0.264				0.774
Yes	2 (3.2%)	1 (1.7%)	(0.0%)		2 (3.5%)	1 (1.7%)	0 (0.0%)	
No	61 (96.9%)	57 (98.3%)	89 (100%)		56 (96.5%)	57 (98.3%)	58 (100.0%)	

^†^*p* < 0.05, when compared with IRE group.

## Data Availability

The data presented in this study is available within the article.

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
