# Peer review of "Safety and Efficacy of Irreversible Electroporation in Locally Advanced Pancreatic Cancer: An Evaluation from a Surgeon’s Perspective"

_cancers, 2022, doi:10.3390/cancers14225677_

Round 1
Reviewer 1 Report
· The Authors reported in the introduction section the statement “Up to date, IRE is the only treatment that can be 72 safely applied to tumors in adjacent to critical anatomical structures”. However, other treatment could be used safety in LAPC and with tumors in adjacent to critical anatomical structures such as Electrochemotherapy.
This, also as preliminary results, was investigated in the recent years as treatment in LAPC. The authors should reported also Electrochemotherapy and also the recent publication in this field:
1: Izzo F, Granata V, Fusco R, D'Alessio V, Petrillo A, Lastoria S, Piccirillo M, Albino V, Belli A, Tafuto S, Avallone A, Patrone R, Palaia R. Clinical Phase I/II Study: Local Disease Control and Survival in Locally Advanced Pancreatic Cancer Treated with Electrochemotherapy. J Clin Med. 2021 Mar 22;10(6):1305. doi: 10.3390/jcm10061305. PMID: 33810058; PMCID: PMC8005134.
2: Granata V, Fusco R, Setola SV, Piccirillo M, Leongito M, Palaia R, Granata F, Lastoria S, Izzo F, Petrillo A. Early radiological assessment of locally advanced pancreatic cancer treated with electrochemotherapy. World J Gastroenterol. 2017 Jul 14;23(26):4767-4778. doi: 10.3748/wjg.v23.i26.4767. PMID: 28765698; PMCID: PMC5514642.
3: Tafuto S, von Arx C, De Divitiis C, Maura CT, Palaia R, Albino V, Fusco R, Membrini M, Petrillo A, Granata V, Izzo F; ENETS Center of Excellence Multidisciplinary Group for Neuroendocrine Tumors in Naples (Italy). Electrochemotherapy as a new approach on pancreatic cancer and on liver metastases. Int J Surg. 2015 Sep;21 Suppl 1:S78-82. doi:10.1016/j.ijsu.2015.04.095. Epub 2015 Jun 27. PMID: 26123385.
4: Granata V, Fusco R, Piccirillo M, Palaia R, Petrillo A, Lastoria S, Izzo F. Electrochemotherapy in locally advanced pancreatic cancer: Preliminary results. Int J Surg. 2015 Jun;18:230-6. doi: 10.1016/j.ijsu.2015.04.055. Epub 2015 Apr 24. PMID: 25917204.
· The authors should clarify the matching procedure in the methods section.
· In the results section the authors reported the statement “A total of 91.38% (52/58) of 205 patients in the VR group underwent R0 resection.”
The percentage value is not correct: 52/58 = 89.66%
· In the Table 2, the authors lost to insert the percentage values for Intra-abdominal infection in IRE group After matching
· In the text the percentage value for intra-abdominal hemorrhage (6.34%) is different from one reported in the Table 2. The correct value of 7.9% reported in the table.
· In the results section the authors reported the statement: “Patients in the IRE group experienced more grade III-V complications, reintervention, and 228 intra-abdominal hemorrhage compared with those in the PS group. Therefore, we have 229 demonstrated that IRE was a safe treatment for patients with LAPC.”
The statement is not clear. How can the authors say that IRE was a safer treatment for patients with LAPC than palliative surgery?
Moreover, this statement “Therefore, we have 229 demonstrated that IRE was a safe treatment for patients with LAPC” should be modified and should be moved in the discussion section.
However, the comparison of complications should be compared with VR group.
· In the figure 1 the authors should modify the axis label reporting for both images: time (month)
· In the discussion section the statement: “We have 293 shown that IRE yielded a significantly longer OS than the other two groups” does not conform to the data reported in the results section. Please edit the statement.
· Also, in the discussion section the authors should report the results of studies in this field of electrochemotherapy as not thermal local ablation alternative treatment.
Reviewer 2 Report
The authors in this original article tried to evaluate the safety and efficacy of IRE against LAPC, and explored its impact on anti-tumor immunity. A retrospective analysis is presented. The study included patients who were assigned to IRE, palliative surgery (PS), and venous resection (VR) groups according to their respective treatments. A one-to-one propensity score matching was performed to compare the incidence of complications and overall survival (OS). Authors concluded that there is survival advantage in IRE-treated patients was attributed to the tumor ablation and immune modulation effects of IRE.
However, the following points have arisen and need to be addressed.
-The included patients apparently don’t belong to LAPC group, since LAPC are inoperable with no feasibility to reconstruct the veins involved according to the definition from latest NCCN guidelines. Therefore, the generalized concept that all patients were LAPC should be redefined.
-patients in PS group were patients either unfit for surgery or declined to have curative resection.
This could attribute to dissimilarities between the groups used in the study. Would it be more convenient and statistically correct if only LAPC cancer were included in PS group? If in this group authors have included also resectable cases, that could be a drawback in their design of study
-Please clarify whether the group comparisons were done as pairwise comparison with ANOVA or kruskal wallis test and if the authors applied any correction (Bonferroni) to deal with type 1 error due to multiple comparisons. The same applies for comparisons made by X2 test.
-define ‘disease complicated by chronic organ insufficiency’ in exclusion criteria’
-propensity score matching should also include either ASA or CCI scores
-the survival rates presented are medial survivals?
-In the IRE group, 17.46% of patients experienced CDC III-V complications, and 2 deaths were observed. Could this be acceptable in the era of neoadjuvant treatment?
Round 2
Reviewer 1 Report
The manuscript has been sufficiently improved to warrant publication in Cancers.
Author Response
We appreciate the reviewer's positive feedback.